



**Remote Quantification of the Trophic Status of Chinese Lakes**
Sijia Li[a], Shiqi Xu[a], Kaishan Song[a], Tiit Kutser[b], Zhidan Wen[a*], Ge Liu[a], Yingxin Shang[a],
Lili Lyu[a], Hui Tao[a], Xiang Wang[a], Lele Zhang[a], Fangfang Chen[a]
[a] *Northeast Institute of Geography and Agroecology, Chinese Academy of Sciences, Changchun*
*130102, China. P. R*
[b] *Estonian Marine Institute, University of Tartu, Mäealuse 14, 12618 Tallinn, Estonia*
*Correspondence* to: Zhidan Wen (wenzhidan@iga.ac.cn)
**Abstract:** Assessing eutrophication in lakes is of key importance, as this parameter
constitutes a major aquatic ecosystem integrity indicator. The trophic state index (*TSI*),
which is widely used to quantify eutrophication, is a universal paradigm in scientific
literature. In this study, a methodological framework is proposed for quantifying and
mapping *TSI* using the Sentinel Multispectral Imager sensor and fieldwork samples. The
first step of the methodology involves the implementation of stepwise multiple
regression analysis of the available *TSI* dataset to find some band ratios, such as
blue/red, green/red, and red/red, which are sensitive to lake *TSI*. Trained with in situ
measured *TSI* and match-up Sentinel images, we established the XGBoost of machine
learning approaches to estimate *TSI*, with good agreement ($R^2$=0.87, slope=0.85) and
fewer errors (MAE= 3.15 and RMSE=4.11). Additionally, we discussed the
transferability and applications of XGBoost in three lake classifications: water quality,
absorption contribution, and reflectance spectra types. We selected the XGBoost to map
*TSI* in 2019-2020 with good quality Sentinel-2 Level-1C images embedded in ESA to
examine the spatiotemporal variations of the lake trophic state. In a large-scale
observation, 10-m *TSI* products from investigated 555 lakes in China facing
eutrophication and unbalanced spatial patterns associated with lake basin characteristics,
climate, and anthropogenic activities. The methodological framework proposed herein
could serve as a useful resource toward a continuous, long-term, and large-scale



monitoring of lake aquatic ecosystems, supporting sustainable water resource
management.

## 1 Introduction

Lakes, as valid sentinels of global or regional responses, are sensitive to
anthropogenic activities and climate change (Mortsch et al., 1996; Quayle et al., 2002;
Tranvik et al., 2009). The commonly used paradigm for studying eco-environmental
monitoring and controlling of lakes is the status of eutrophication (Carlson, 1977). It is
a combination of light, heat, hydrodynamics, and nutrients, such as nitrogen and
phosphorus, which occurs through a series of biological, chemical, and physical
processes of lakes. As a result of eutrophication, nutrient loading and productivity grow
sharply, and even hypoxia and frequent outbreaks of harmful algal blooms are likely to
produce toxins (Paerl et al., 2008, 2011). These processes can cause serious degradation
of water quality and are detrimental to the ecosystem services functionality of lakes and
reliable supply of drinking water (OECO, 1982). Once the eutrophication phenomenon
becomes intense, ecological imbalances generally follow (Smith et al., 2006). Hence,
knowledge of the process of eutrophication can provide us with an understanding of the
structure and function of lake ecosystems that give rise to environmental changes. We
can then predict future trends and develop appropriate mitigation strategies.
Several lakes experience eutrophication processes because of excessive nutrient
enrichment (Lund, 1967; Smith et al., 1999; Wetzel, 2001). At the global scale, 63.1%
of lakes larger than 25 $km^2$ are eutrophic and 54% of Asian lakes (Wang et al., 2018), as
well as 53% of European lakes (ILEC et al., 1994). Lake eutrophication has become a
global water quality issue affecting most freshwater ecosystems (Matthews, 2014).
Currently, many pollutions control measures and management strategies have been
implemented that are specific to individual lakes or to lakes, in general (USEPA, 2002).



However, there is still insufficient information to address lake eutrophication related to
environmental disturbances or changes. Realization of lake eutrophication has been a
serious situation for some lakes; therefore, we provided some reasons to suggest the
need for large-scale research. First, different environmental factors control the trophic
status of lakes at local and multiple scales (e.g., Wiley et al., 1997). Specifically, biotic
factors may dominate the eutrophic state of individual lakes, and we can understand the
mechanism processes by lake-specific sampling. In contrast, abiotic factors and their
linkages are pivotal factors that determine lake biogeochemistry at multiple scales (Sass
et al., 2007). It is often necessary to study a number of lakes with different
characteristics and catchments to understand the mechanisms of spatio-temporal
patterns. Therefore, an up-scaling study of trophic status is required to understand the
evolution prospects of lakes in response to changes in global and regional environments.
Second, multi-year environmental and climatic conditions require long-term field
studies and observations to understand the temporal pattern in important trophic status
processes. In addition, relatively large datasets are needed considering the spatial extent
because environmental factors are integrated to determine the trophic status of lakes. It
can promote data organization and enable us to address an emergency and establish
scientific measures for water resource management (Cunha et al., 2013; Smith and
Schindler, 2009). Thus, eutrophication should be rapidly assessed using easy-to-analyze
indices and enforcement methods for large-scale and high-frequency applications.

73        Evaluating the trophic state of lakes has been an important topic for decades

(Carlson, 1977; Smith and Schindler 2009). The traditional method uses chlorophyll-a,
transparency, nutrients, and other variables as water quality indicators by field in situ
sampling and laboratory measurements (Rodhe, 1969). Subsequently, Carlson (1977)
introduced a numerical *TSI* that should have replaced descriptive values like
"oligotrophic," "mesotrophic," or "eutrophic". The replacement has not occurred, but
the *TSI* proposed by Carlson is a common method to determine the trophic state level of
aquatic environments (Aizaki et al., 1981). The traditional method for calculating *TSI* is
based on collected in situ data. The sampling itself and subsequent laboratory
measurements are labor-intensive and expensive, often also logistically difficult to
perform. This limits our capability to monitor hundreds or thousands of lakes for
eutrophication, not speaking about the majority of 117 million of lakes on Earth
(Verpoorter et al. 2014). Moreover, the TSI calculated for one or a few discrete samples
do not represent spatial distribution of TSI within (especially larger) lakes. This could
limit the large-scale assessment of eutrophication as well as the understanding of
biogeochemical cycles.

89        Satellite remote sensing is a useful tool for monitoring inland waters (Palmer et

al 2015). Ocean water-color sensors, such as Medium Resolution Imaging Spectrometer
(MERIS) or Ocean and Land Colour Instrument (OLCI) have too low spatial resolution
(300 m) for majority of lakes on Earth. Land remote sensing seosor like Landsat
Operational Land Imager (OLI), Sentinel-2 Multispectral Imager (MSI; 10-60 m) and
Satellite pour l'Observation de la Terre (SPOT) with high spatial resolution (5–30 m) are
not designed for water remote sensing (lack critical spectral bands, SNR is not sufficient
for water, etc.). Compared to OLI and SPOT sensors, MSI has a more adequate
radiometric resolution (12-bits) and 13 spectral bands, including four visible and SWIR
channels (Drusch et al., 2012). Inland water TSI has been produced for large lakes using
MODIS sensor (Wang et al 2018). However, this study is for more than 2000 large lakes
(due to the spatial resolution of the sensor) while there are 117 million of lakes on Earth
(Verpoorter et al. 2014). The Copernicus Land Monitoring Service has started to
produce TSI for lakes large enough to be mapped with 100 m pixel size using Sentine-2


MSI. However, this product is available only for Europe and some parts of Africa.
Instead of individual parameters, several studies (e.g., Morel and Prieur, 1977;
Gurlin et al., 2011; Huang et al., 2014; Sass et al., 2007; Thiemann & Kaufmann, 2000;
Yin et al. 2018) have also provided empirical relationships expressed as band
combinations or baseline methods to acquire Chl-a, Secchi or nutrients related to
potential *TSI* calculations in regional lakes. However, the accuracy of these empirical
relationships for transferring knowledge from some representative lakes to large-scale
lake groups is limited by large uncertainties (i.e., in areas with different water quality
concentrations and atmospheric component influences, fewer lakes can be used with
more heterogeneous influences and uniform algorithms) (Oliver et al., 2017).
Considering the requirement of a uniform and universal relationship to quantify the
trophic status of lakes, an alternative method using high-frequency and spatial
resolution of the sensor is a significant challenge. Recently, technological developments,
such as machine learning algorithms, have allowed the usage of remotely sensed
imagery to successfully investigate water quality parameters using artificial intelligence
(Reichstein et al., 2019; Pahlevan et al., 2020; Cao et al., 2020). The potential
application and development of machine learning for remote quantification of water
quality is attributed to the following advantages: requirement of little prior knowledge,
rich features can be captured, and robust relationships can be obtained. These processes
avoid bias and uncertainty from the regional environmental background as well as
complications due to atmospheric components of traditional remote sensing-derived
relationships over large-scale, i.e. for multiple lakes. Given the novel application of
remote sensing and machine learning, this is a gap to fill for large-scale research of
monitoring trophic states.
Environmental issues fueled by rapid economic growth in China have significantly



increased in the last three decades. Lake eutrophication is a serious issue, with large
variability in terms of trophic status and optical properties. However, most studies (Jin,
2003, 2005; Fragoso et al., 2011; Huang et al., 2014) have addressed eutrophication
concerns in only a single lake or two lakes since the 1990s. It is acknowledged that a
rapidly growing economy and anthropogenic activities (e.g., elevated nutrient loading
and increasing air pollution) accelerate the aging process of lakes (Wu et al., 2011; Shi
et al., 2020). Therefore, it is critical to objectively assess the trophic status and pay
attention to protect the aquatic environment. We aim to provide a robust machine
learning algorithm and remote sensing flowchart from simultaneously retrieved *TSI*
over a wide range of bio-optical compositions in different lakes. The objectives of our
study were to: (1) examine biogeochemical parameters and assess trophic status, (2)
calibrate and validate the *TSI* model using different machining learning algorithms from
MSI-imagery derived remote sensing reflectance spectra *(Rrs)*, with different lake
classifications; and (3) quantify and map the trophic status of typical 555 lakes in five
Chinese limnetic regions.
**2 Materials and methods**
**2.1 *Study area and sampling process***
China is located in the east of Asia with a land area of 9,600,000 square kilometers
and a population of over 1.4 billion. The terrain of China descends from west to east in
three steps. Due to a vast territory span, this country has diverse climatic, geographical,
and geological conditions. There are 2,693 natural lakes (with area >1.0 km$^2$) that are
distributed in China (Ma et al., 2011). Protection and sustainable management of these
lakes have been priorities, considering the degradation of water quality over several
decades. In this study, a total of 45 lakes were visited and 431 samples were collected in
early April 2016 to late October 2019 (Table S1 and Fig. 1), which was the highest





productive season, as identified by Carlson's *TSI* model. These datasets were analyzed
and published in (Li et al., 2021; Song &Li et al., 2019; Song et al., 2020). Our lake
dataset was collected from various types of lakes across China, and efforts were made to
examine lake trophic status from a wide range of water quality parameters, lake sizes
(0.5 to 4, 256 km$^2$), lake elevation (10 to 4, 525 m), and climatic zones (Song and Li et
al., 2019). In the field, some lakes were sampled in the middle while others were
sampled at multiple locations evenly distributed over the lake. The water samples were
collected approximately 0.5 m below the surface, and then stored in 1 L amber HDPE
bottles and kept in a portable refrigerator (4°C) before being transported to the
laboratory. During the sampling process, the Secchi disk depth (SDD, m) was measured
using a black-and-white Secchi disk. The pH and electrical conductivity (EC, μs cm$^{-1}$)
were recorded using a portable multi-parameter water quality analyzer (YSI 6600, 170
U.S).

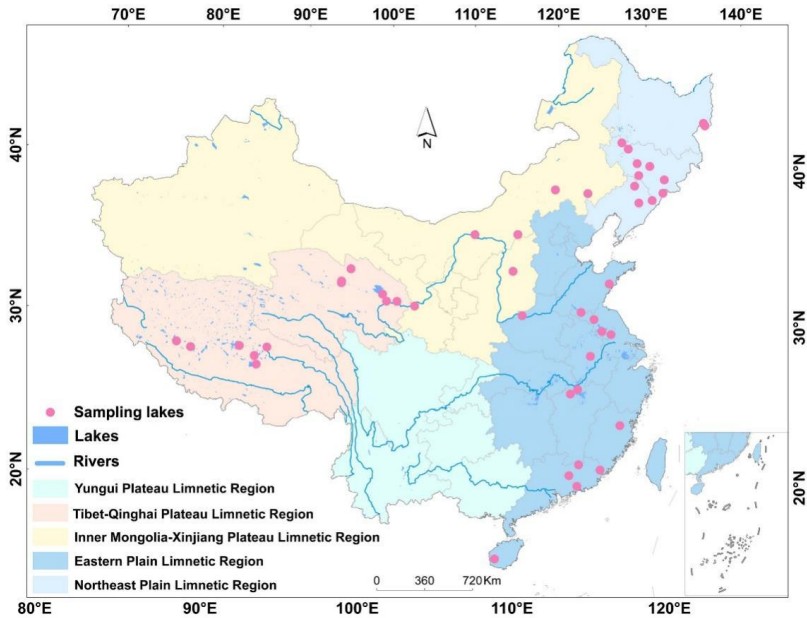


**Figure 1:Location of lake sites.**



## 2.2 Laboratory analysis

A transferred portion of each bulk water sample was immediately filtered with 0.45-μm pore size Whatman cellulose acetate membrane filters in the laboratory. It is to be noted that some remote Tibet and Qinghai lake samples had to be filtered during fieldwork. Chlorophyll-a (Chl-a) was extracted from the filters using a 90 % buffered acetone solution at 4° C under 24 h dark conditions. According to the SCOR-UNESCO equations (Jeffrey and Humphrey, 1975), the concentration of Chl-a (μg $L^{-1}$) was determined using a UV-2600PC spectrophotometer at 750 nm, 663 nm, 645 nm, and 630 nm. Dissolved organic carbon (mg $L^{-1}$) concentrations were determined using a total organic carbon analyzer. Total nitrogen (TN) and total phosphorus (TP) concentrations (mg $L^{-1}$) were measured using a continuous flow analyzer (SKALAR, San Plus System, the Netherlands) using a standard procedure (APHA/AWWA/WEF, 1998). In addition, total suspended matter (TSM, mg $L^{-1}$) concentrations were obtained gravimetrically using pre-combusted 0.7-μm pore size Whatman GF/F filters. All preprocesses (e.g., filtration and concentration quantification) of all water samples were undertaken within two days in the laboratory. The procedures are provided in detail in Li et al. (2021).

The bulk samples were again filtered through a 0.7-μm pore size glass fiber membrane (Whatman, GF/F 1825-047) to retain particulate matter. The water from particulate matter measurements was then filtered through a 0.22-μm pore size polycarbonate membrane (Whatman, 110606) in order to measure chromophoric dissolved organic matter (CDOM) absorption of each sample. According to the quantitative membrane filter technique (Cleveland and Weidemann, 1993), the light absorption of total particulate matter $a_p(\lambda)$ can be separated into phytoplankton pigment absorption $a_{ph}(\lambda)$, non-algal particles $a_d(\lambda)$, and CDOM absorption $a_{CDOM}(\lambda)$. The optical



density (OD) of the particulate matter retained in the filters was measured using a
UV-2600PC spectrophotometer at 380–800 nm, with a blank membrane as a reference
at 380–800 nm. The filters were then bleached using a sodium hypochlorite solution to
remove phytoplankton pigment and measured again using a spectrophotometer. Finally,
the phytoplankton pigment absorption $a_{ph}(\lambda)$ was calculated by subtracting $a_d(\lambda)$ from
the total particulate matter $a_p(\lambda)$. The absorption coefficients of the optical active
substance (OACs) were calculated according to Song et al. (2013).

**2.3 Trophic status assessment of lakes**

Several studies have proposed different indices of the lake trophic state (Aizaki et al.,
1981; Carlson, 1977). Carlson's trophic state index used five variables, such as Chl-a,
TP, TN, SDD, and chemical oxygen demand (COD), to characterize the trophic state.
However, there are no optical characteristics for TN, TP and COD to manifest in
changes of remote sensing reflectance, which may bring more uncertainties or errors.
Thus, Chl-a, TP, and SDD were selected to assess the trophic status according to the
modified Carlson's trophic state index (*TSI*). The *TSI* can be calculated using individual
*TSI*$_M$(Chl-a), *TSI*$_M$(SDD), and *TSI*$_M$(TP) using the following equations:

$$TSI_M(Chl\text{-}a) = 10 \times \left( 2.46 + \frac{\ln Chl - a}{\ln 2.5} \right) \tag{1}$$

$$TSI_M(SDD) = 10 \times \left( 2.46 + \frac{3.69 - 1.52 \times \ln SDD}{\ln 2.5} \right) \tag{2}$$

$$TSI_M(TP) = 10 \times \left( 2.46 + \frac{6.71 + 1.15 \times \ln(TP)}{\ln 2.5} \right) \tag{3}$$

$$TSI = 0.54 \times TSI_M(Chl\text{-}a) + 0.297 \times TSI_M(SDD) + 0.163 \times TSI_M(TP) \tag{4}$$

Where, the *TSI* below 30 correspond to oligotrophic waters, above 50 are eutrophic and
TSI between 30 and 50 in mesotrophic (Carlson, 1977).

**2.4 Muti-Spectral Instrument imagery and atmospheric correction**

Sentinel-2A/B MSI imagery was acquired from the Copernicus Open Access Hub
of the European Space Agency. Altogether, 210 scenes of cloud-free Level-1C images
covering the lakes were downloaded with a time window of ±7 days from in situ
measurements. The Case 2 Regional Coast Color processor (C2RCC) was used to
remove atmospheric effects. An average of 3×3-pixels centered at each in situ sampling
station was used in the further analysis. All the processes were performed using the
Sentinel Application Platform (SNAP) version 7.0.0. A flowchart of the process is
shown in Fig. 2.

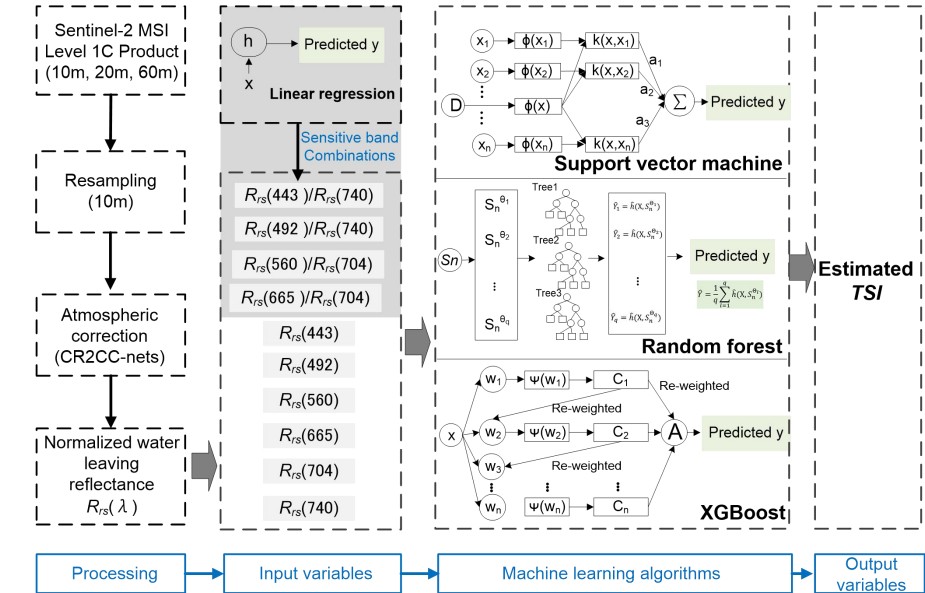


**Figure 2 : Workflow of the Sentinel-2 MSI data and machine learning algorithms**
**for estimating *TSI***

**2.5 Machine learning algorithms**
As a branch of artificial intelligence, the application of machine learning is
growing in the field. Machine learning can automatically analyze huge chunks of data,
develop optimal models, generalize algorithms, and make predictions. These approaches
have been applied in a variety of eco-environmental and remote sensing fields
(Mountrakis et al., 2011; Pahlevan et al., 2019). Hence, we employed four


representative machine learning algorithms, namely linear regression (LR), support
vector machine (SVM), XGBoost (XGB), and random forest (RF) (Supplementary data,
methods), to establish a *TSI* model. To strengthen the robustness, band combinations
sensitive to *TSI* were determined by LR (Fig. 2), and were added to the procedure of
machine learning algorithms as input variables. Subsequently, the output variable was
the predicted *TSI*. The in situ measured samples were then randomly divided into a
calibration dataset (70%, 287 lake samples) and validation dataset (30%, 144 lake
samples) using MATLAB software. The *TSI* modeling procedure considering machine
learning and Multiple Linear Regression (MLR) was processed using the R software.
**2.6 Classifications of lakes**
In order to provide further feasibility for the application and availability of the *TSI*
model, the in situ measured samples were classified in three ways (Fig. 3):
a) based on water quality: Salinity classification referred to the threshold value of
electrical conductivity (named EC, EC=1000 $\mu$S cm$^{-1}$) (Duarte et al., 2008), following
which the lakes were divided into brackish lakes ($N$=100 samples) and fresh water lakes
($N$=331 samples). Dissolved organic carbon (DOC) in global lake water classification
referred to the volume weighted averaged DOC level of global lakes (3.88 mg L$^{-1}$)
according to Toming et al., (2020), following which lakes were divided into high DOC
lake ($N$=224 samples) and low DOC lake ($N$=207 samples).
b) based on optical absorption contribution: Optical absorption classification
referred to Prieur and Sathyendranath (1981), where the total light absorption of water
can be separated from phytoplankton pigment absorption, non-algal particles, and
CDOM absorption, respectively. The relative percentage of absorption contribution of
OACs can be divided into phytoplankton-type (Phy-type) lakes ($N$=54 samples),
non-algal particles-type (NAP-type) lakes ($N$=109 samples), CDOM-type lakes ($N$=177





samples), and mix-type lakes (*N*=91 samples).
c) based on reflectance spectra: In order to discern the different optical
characteristics of lakes, the derived MSI reflectance was clustered using the k-means
clustering approach with a gap statistic (Neil et al., 2018). We identified 431 MSI
reflectance $Rrs(\lambda)$ spectra for three branches (Table S3), and the $Rrs(\lambda)$ spectra are
shown in Fig.3.

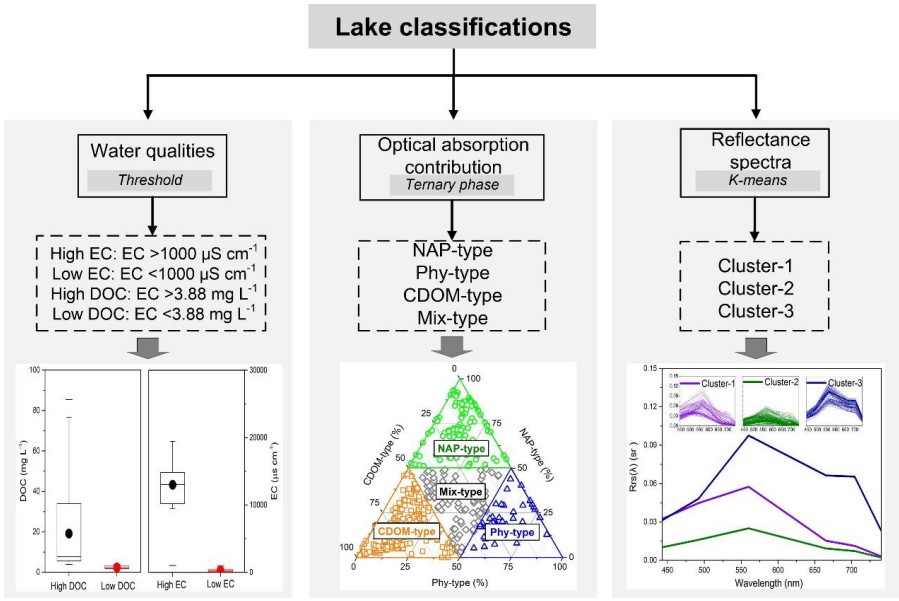


**Figure 3:Lake classifications considering three ways, i.e., water quality, optical**
**absorption contribution and reflectance spectra. ANOVA analysis was conducted**
**in different classifications (*p*<0.001) (Table S3).**
**2.7 Statistical analyses and accuracy assessment**
Statistical analysis, including descriptive statistics, correlation (*r*), regression ($R^2$),
and ANOVA analyses, were implemented with Statistical Program for Social Science
software (version 16.0; SPSS, Chicago, IL, USA). Correlation and regression analyses
were used to examine the relationships between the water quality parameters and
absorption coefficients of OACs as well as the *TSI* model calibration and validation.
The differences in trophic status, EC classification, DOC classification, absorption

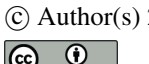



coefficients of OAC classification, and MSI reflectance spectra classification for *TSI*
model validation were assessed using one-way ANOVA. The significance level was set
at $p<0.05^*$. The mean normalized error (MAE) and root mean square error (RMSE)
were used to assess the performance of the *TSI* model (Supplementary data, accuracy
assessment).
**3 Results**
**3.1 Aquatic environmental scenery**
The water quality and bio-optical properties covered a wide range of nutrient
compositions, transparencies, and trophic states, revealing different geographical
environmental scenery (Tables S1 and S2-4). The EC and DOC concentration showed
high variability, ranging for example, from 3345.31 μs cm$^{-1}$ (TuoSu, TS20) in
Tibet-Qinghai region to 0.17 μs cm$^{-1}$ (Qingnian, QN2) in Northeast region. For the
water quality parameters to characterize *TSI*, the Chl-a concentration ranged from 0.12
to 100.22 μg L$^{-1}$, with the highest value recorded in TaiPingChi (TPC5) and the lowest
value in NamoCo (NMC36). The range of TP was from 0.003 mg L$^{-1}$ (Erlong, EL8) to
2.17 mg L$^{-1}$ (Dali, DL7), and SDD ranged from 0.17 m (Chalhu, CH32) to 9.47 m
(NMC36) for surveyed lakes, respectively. Overall, the maximum values of EC, DOC,
turbidity, Chl-a, TSM, and SDD were 196782.35, 948.4, 723.3, 770.92, 614.58, and
55.71 fold greater than the minimum values, respectively, indicating that our dataset
was representative of diverse water qualities.
Lake samples were grouped into different classifications based on water quality
(e.g., EC and DOC), optical absorption contribution, and reflectance spectra (Table 1
and Fig. 3). The results indicated that all water qualities showed significant differences
($p<0.05$) under different lake classifications. For example, brackish lakes showed higher
average values of SDD, TP, DOC, and optical attributions of OAC values than those of


fresh water lakes, but the turbidity, Chl-a, and TSM concentrations were lower. Lakes
equipped with low DOC levels had a low average value of SDD than that of lakes with
high DOC levels. NAP-type lakes exhibited the highest average Chl-a and DOC values,
whereas Phy-type lakes had the highest average turbidity and TSM values, and the
highest average SDD and TP values were recorded in CDOM-type and Mix-type lakes,
respectively. For reflectance spectra classifications (Fig. 3), the highest average EC,
SDD, and DOC were recorded in cluster-1 lakes, the highest average turbidity and TP
was shown in cluster-3 lakes and the highest average TSM was found in cluster-2 lakes.






**Table1 (a)** Averaged values (Avg.) of water quality and bio-optical properties considering lake classifications and **(b)** ANOVA analysis (*F* value) among them

**(a)**

| Lake classifications | | $N$ | EC | Turbidity | SDD | Chl-a | TP | DOC | TSM | $a_{ph}(440)$ | $a_d(440)$ | $a_{CDOM}(440)$ |
|---|---|---|---|---|---|---|---|---|---|---|---|---|
| Water quality | Brackish | 100 | 12986.28 | 8.83 | 2.21 | 4.18 | 0.45 | 33.31 | 8.42 | 0.23 | 0.27 | 0.42 |
| | Fresh | 331 | 302.39 | 21.75 | 1.43 | 8.58 | 0.07 | 4.28 | 19.52 | 0.56 | 1.13 | 0.57 |
| | High DOC | 224 | 5988.93 | 23.90 | 1.39 | 10.42 | 0.25 | 19.07 | 21.50 | 0.68 | 1.14 | 0.65 |
| | Low DOC | 207 | 276.19 | 12.45 | 1.85 | 4.46 | 0.06 | 2.29 | 11.98 | 0.27 | 0.71 | 0.41 |
| Optical absorption contribution | NAP-type | 54 | 5156.02 | 11.28 | 1.58 | 14.26 | 0.09 | 18.75 | 15.99 | 1.29 | 0.41 | 0.55 |
| | Phy-type | 109 | 825.48 | 43.28 | 0.65 | 6.85 | 0.10 | 4.75 | 37.18 | 0.46 | 2.74 | 0.49 |
| | CDOM-type | 177 | 4081.96 | 4.44 | 2.43 | 3.64 | 0.13 | 9.70 | 4.99 | 0.13 | 0.15 | 0.51 |
| | Mix-type | 91 | 3424.07 | 19.40 | 1.17 | 12.05 | 0.34 | 16.48 | 16.22 | 0.70 | 0.60 | 0.62 |
| Reflectance spectra | Cluster-1 | 87 | 6948.28 | 4.46 | 2.38 | 2.64 | 0.08 | 17.92 | 5.76 | 0.26 | 0.17 | 0.28 |
| | Cluster-2 | 215 | 2728.71 | 6.18 | 2.05 | 8.57 | 0.07 | 7.18 | 5.81 | 0.35 | 0.36 | 0.52 |
| | Cluster-3 | 129 | 1626.05 | 46.68 | 0.36 | 9.19 | 0.37 | 12.73 | 42.59 | 0.84 | 2.39 | 0.73 |

**(b)**

| Lake classifications | | $N$ | EC | Turbidity | SDD | Chl-a | TP | DOC | TSM | $a_{ph}(440)$ | $a_d(440)$ | $a_{CDOM}(440)$ |
|---|---|---|---|---|---|---|---|---|---|---|---|---|
| Water quality | Brackish | 100 | - | 18.7** | 21.8** | 12.0** | 68.9** | 486.5** | 20.4** | 16.6** | 29.8** | 9.6* |
| | Fresh | 331 | | | | | | | | | | |
| | High DOC | 224 | 93.8** | 19.8** | 10.0** | 32.2** | 23.3** | - | 21.0** | 38.0* | 10.0* | 39.3** |
| | Low DOC | 207 | | | | | | | | | | |
| Optical absorption contribution | NAP-type | 54 | 7.4** | 71.6** | 46.0** | 21.0** | 7.1** | 13.5** | 73.0** | - | - | - |
| | Phy-type | 109 | | | | | | | | | | |
| | CDOM-type | 177 | | | | | | | | | | |
| | Mix-type | 91 | | | | | | | | | | |
| Reflectance spectra | Cluster-1 | 87 | 220.9** | 17.9** | 25.2** | 312.7** | 11.0** | 18.5** | 18.9** | 26.1** | 171.4** | 33.5** |
| | Cluster-2 | 215 | | | | | | | | | | |
| | Cluster-3 | 129 | | | | | | | | | | |

The unit of TN, TP, DOC and TSM is mg L$^{-1}$; EC is µs cm$^{-1}$; Chl-a is µg L$^{-1}$; turbidity is NTU (nephelometric turbidity unit). Significance levels are reported as significant (noted with *, $0.05 > p > 0.01$) or highly significant (noted with **, $p < 0.01$).






## 3.2 Trophic status assessment


The trophic status of 45 lakes across China, from where in situ samples were
collected, was evaluated (Fig. 4a). Our results showed that there were 13 oligotrophic
(3.02 %), 199 mesotrophic (46.17 %), and 219 eutrophic (50.81 %) samples. Because
our samples were collected in different seasons and eutrophication is time-dependent,
the *TSI* values of samples within a lake were averaged. It can be shown that only five
lakes accounting for 11.1% of investigated lakes were characterized with an
oligotrophic status, 17 lakes accounting for 37.8 % were mesotrophic, and 23 lakes
accounting for 51.1 % were characterized with eutrophic status. These eutrophic lakes
were distributed in the eastern region of China (Fig. 4b), and were associated with a
highly concentrated human population and economic development. Moreover, the
ANOVA results showed that the *TSI* of lake samples were significantly different
considering lake classifications (Fig. 4c, and d).


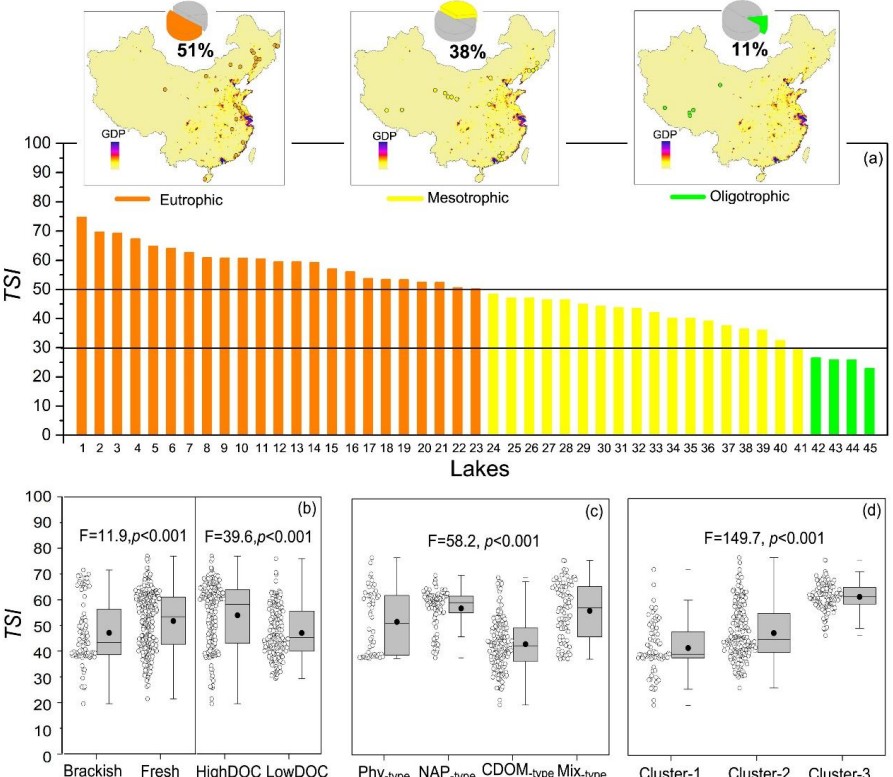

**Figure 4: (a) is the averaged *TSI* in collected samples from lakes across China and their spatial distribution. The number of lakes can be found in TableS1. The box plots of *TSI* at different classifications of water quality (b), optical absorption contribution types (c) and reflectance spectra (d). The balls beside the boxes are the lake samples, and the black balls in the boxes represent the mean values. The horizontal edges of the boxes denote the 25th and 75th percentiles; the whiskers denote the 10th and 90th percentiles.**

**3.3 Calibration and validation of *TSI* model**

In this section, multiple linear regression was used to identify significantly sensitive spectral variables related to *TSI* (Table 2 and Fig. 2). Of the band combinations validated in the study (*N*=144), the blue/red [*Rrs*(443)/*Rrs*(740), *Rrs*(492)/*Rrs*(740)], and green/red [*Rrs*(560)/*Rrs*(704), *Rrs*(665)/*Rrs*(704)] band ratios showed a good regression coefficient ($R^2$>0.59) with *TSI*. These band combinations provided certain sensitive spectral variables that responded to the lake eutrophic status. Hence, to



strengthen the robustness of the three machine learning models, the blue/red and
green/red combinations above were considered as the input variables as well as six
spectral variables ($Rrs(\lambda)$ at 443, 492, 560, 665, 709, and 740 nm). Likewise, the output
variables were estimated using *TSI* to examine the performances (Fig. 5). The results
showed that when XGBoost was applied to the validation data (*N*=144), the
performance of the model was excellent ($R^2$=0.87, slope=0.85) with low errors (MAE=
3.15, RMSE=4.11). The support vector machine ($R^2$=0.71, slope=0.77, MAE=4.67,
RMSE=6.11) and random forest ($R^2$=0.85, slope=0.84, MAE=3.31, RMSE=4.34)
models also showed significant performance. These results demonstrate the potential of
using XGBoost by considering band combinations to derive *TSI* from Sentinel products.

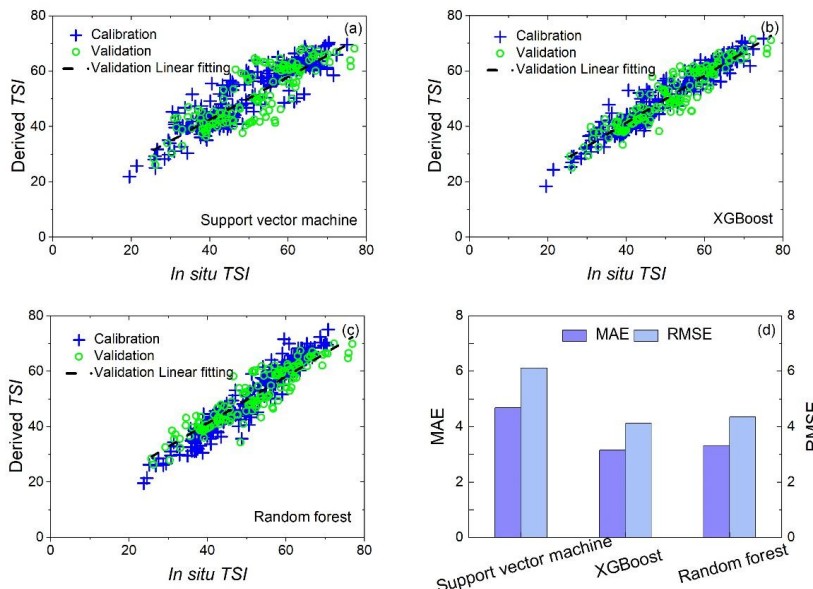


**Figure 5: Relationships between in situ and derived TSI for both model training**
**and testing samples by support vector machine (a), XGBoost (b) and random**
**forest (c), as well as their errors (d).**










**Table 2** Multiple linear regression between measured- and estimated- *TSI* from the MSI spectral bands after using CR2CC processor

| Band combinations | Datasets | N | Fitting equation | $R^2$ | Errors | Plots figures |
|---|---|---|---|---|---|---|
| Band 1 / Band 6 (Blue/Red) | Calibration | 287 | $TSI = -8.51\ln[Rrs(B1)/Rrs(B6)] + 63.47$ | 0.76 | MAE = 6.45, RMSE = 5.85 | |
| | Validation | 144 | $TSI_{derived} = 0.73 \times TSI_{in\ situ} + 11.868$ | 0.61 | MAE = 6.26, RMSE =7.48 | |
| Band 2/ Band 6 (Blue/Red) | Calibration | 287 | $TSI = -8.87\ln[Rrs(B2)/Rrs(B6)] + 67.91$ | 0.77 | MAE = 4.57, RMSE = 5.74 | |
| | Validation | 144 | $TSI_{derived} = 0.74 \times TSI_{in\ situ} + 11.751$ | 0.60 | MAE = 6.32, RMSE =7.57 | |
| Band 3/ Band 5 (Green/Red) | Calibration | 287 | $TSI = -13.63\ln[Rrs(B3)/Rrs(B5)] + 67.26$ | 0.77 | MAE = 4.55, RMSE = 5.70 | |
| | Validation | 144 | $TSI_{derived} = 0.72 \times TSI_{in\ situ} + 12.44$ | 0.59 | MAE = 6.39, RMSE = 7.66 | |
| Band 4, Band 5 (Red/Red) | Calibration | 287 | $TSI = -44.15 \times [Rrs(B4)/Rrs(B5)] + 108$ | 0.80 | MAE = 4.39, RMSE = 5.43 | |
| | Validation | 144 | $TSI_{derived} = 0.72 \times TSI_{in\ situ} + 12.32$ | 0.59 | MAE = 6.85, RMSE = 7.94 | |



### 3.4 *TSI* model application to lake classifications

The *TSI* model calculated by XGBoost was assessed by comparing derived and in situ *TSI* considering different lake classifications (Fig. 6). We aimed to provide a universal *TSI* model and evaluate its feasibility in different aquatic environments. Significant agreement (slope>0.91, $R^2$>0.91) between derived and in situ *TSI* was observed in lakes with high DOC levels (DOC>3.88 mg $L^{-1}$) and EC values (EC>1000 μS $cm^{-1}$), with low errors. For lakes classified by different absorption contributions, the NAP-type (slope=0.98, $R^2$=0.88) and Phy-type (slope=0.82, $R^2$=0.92) samples generally showed a positive derived performance than those of Phy-type, CDOM-type, and Mix-type, respectively. In addition, a significant relationship between derived and in situ *TSI* can be described for lakes with cluster-1 reflectance spectra, with slope=0.91, $R^2$=0.87, RMSE=2.87, and MAE=2.29.



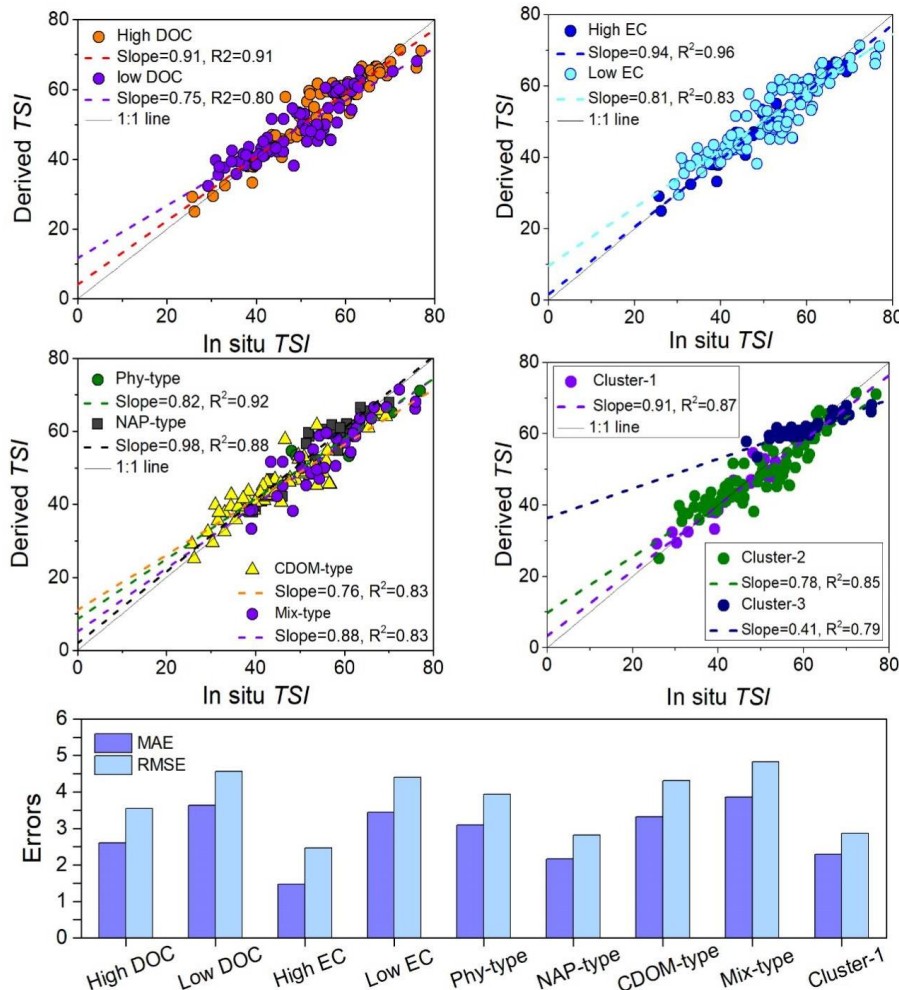

**Figure 6:Scatter plots of derived- and in situ- TSI by XGBoost for validation samples (*N*=144) according to lake classifications, such as water quality (DOC and EC) (a-b), absorption contribution (c), reflectance spectra(d) with the 1:1 line (red solid) and errors (e).**

## 3.5 Spatio-temporal patterns of trophic states in a large-scale overview

Previous studies have demonstrated that some lakes disappeared or increased numbers recently according to statistics from Ma et al. (2011). Thus, we selected some representative lakes (*N*=555) to qualify spatiotemporal trophic states using the XGBoost algorithm. According to the different geographic and limnological types in China, lakes

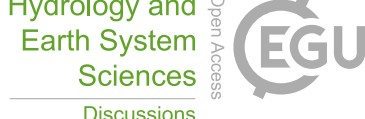

were divided into five limnetic regions (Wang and Dou 1998, Early National
Investigation): Eastern Plain Limnetic Region (EPLR, $N$=123), Northeast Plain
Limnetic Region (NPLR, $N$=37), Inner Mongolia-Xinjiang Plateau Limnetic Region
(IMXPLR, $N$=56), Yungui Plateau Limnetic Region (YGPLR, $N$=15), and
Tibet-Qinghai Plateau Limnetic Region (TQPLR, $N$=324) (Fig. 1 and Supplementary
data). In general, there were significant seasonal variations in eutrophic state for lakes
from the EPLR (F=39.56, $p$<0.001) and TQPLR (F=5.0, $p$<0.05) (Fig. 7). The eutrophic
lakes dominated the proportions of the investigated lakes in the EPLR (93.5 %),
followed by the NPLR (89.2 %) and YGPLR (86.7 %). In comparison, most
mesotrophic and oligotrophic lakes were distributed in the TQPLR. The spatio-temporal
patterns of trophic states in lakes were related to lake basin characteristics, climate, and
anthropogenic activities.

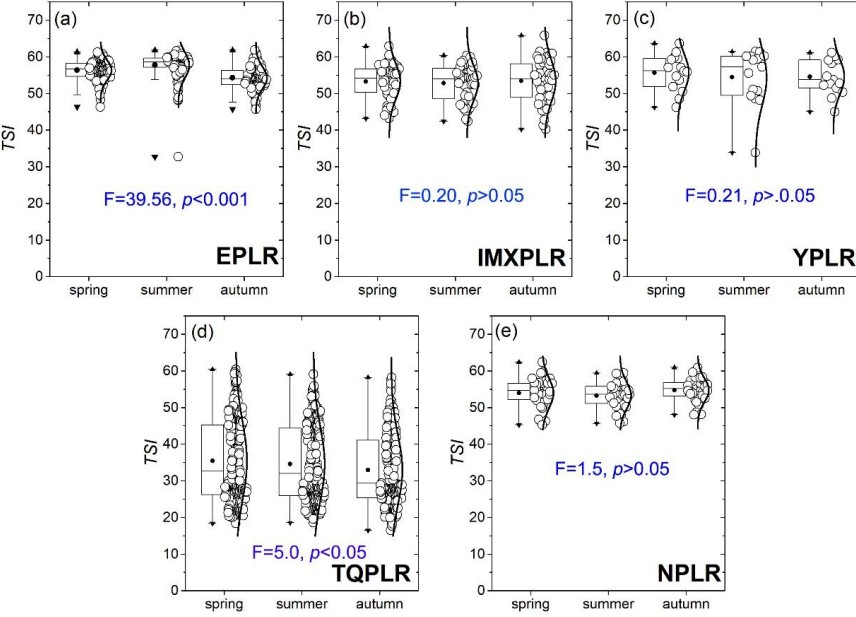


**Figure 7 : Box plots of TSI derived from XGBoost model in investigated lakes**
**from the five limnetic regions (Wang & Dou 1998), i.e., (a) EPLR, (b) IMXPLR, (c)**
**YPLR, (d) TQPLR and (e) NPLR. The black line and balls in the boxes represent**





**the median and mean values, respectively. The horizontal edges of the boxes denote**
**the 25th and 75th percentiles; the whiskers denote the 10th and 90th percentiles.**

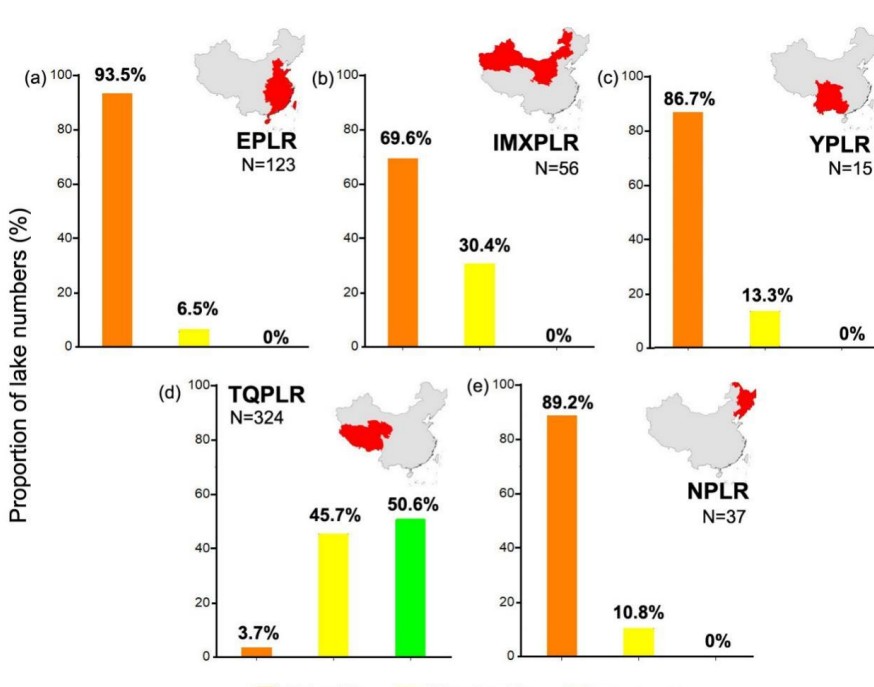


**Figure 8 : The proportions of lake numbers (%) for different trophic state in the**
**five limnetic regions (Wang & Dou 1998), i.e., (a) EPLR, (b) IMXPLR, (c) YPLR,**
**(d) TQPLR and (e) NPLR. N represents the lake numbers.**

**4 Discussion**
**4.1 Remote-sensed and machine-learning-based *TSI* model**
Traditional approaches to quantitatively characterize the trophic status rely on field
measurements of trophic parameters, for example, Chl-a, nutrients, and SDD, to
calculate the *TSI* (Carlson, 1977). It is difficult and costly to make field measurements
in lakes in remote locations. The *TSI* calculation does not need all of these trophic
parameters but just one, for example, Chl-a (Thiemann and Kaufmann, 2000), SDD
(Olmanson et al., 2008; Song et al., 2020), TP (Kutser et al., 1995) and total absorption
coefficients (Lee et al., 1999; Shi et al., 2019), etc. There have been many lake studies


(Chl-a and SDD, Sheela et al., 2011; Chl-a, SDD and TP, Song et al., 2012) where two
or three water quality parameters were mapped, which would allow to subsequently
gather them to calculate comprehensive *TSI*. Although these studies provided the
potential to evaluate the trophic status of lakes, *TSI* is a synthetic indicator that is
affected by biological, physical, and chemical factors that co-vary in most instances.
Huang et al. (2014) also tried to derive *TSI* using remote sensing spectrum reflectance,
but the accuracy was not completely usable. It shows that variability in remote sensing
estimates of the TSI are not bad.

425        With advances in artificial intelligence technology and the increasing use of

computer applications in recent years, machine learning has become a useful tool for
monitoring aquatic environments by remote sensing (Mountrakis et al., 2011). It allows
us to develop and evaluate a machine-learning-based *TSI* model that addresses quality
and accuracy problems more effectively (Li et al., 2021). Hence, we propose a new
approach to directly characterize the trophic status and accurately reflect spatial
variations in this study, but should also be conveniently available for the different lake
classifications (Figs. 5, 6). Using machine learning algorithms, in order to improve the
robustness and applicability of the *TSI* model, a sufficient database of trophic state
parameters (*N*=431) was collected from lakes with different biogeochemical
characteristics, such as water quality, absorption contributions of different optically
active substances, and reflectance spectra (Table1). We first used B1-B6 reflectance as
input variables of machine learning algorithms, and XGBoost showed a significant
performance with $R^2$ and a slope of 0.85 (Fig. S1). The support vector machine and
random forest did not produce the sufficient performance. There was no optical
response bands or appropriate band ratios for *TSI*. We thus used a multiple linear
regression to find some suitable sensitive band combinations responding to the *TSI*,



which made it possible to develop a robust machine-learning-based *TSI* model. It is
important to note that the blue/red [$Rrs$(443)/$Rrs$(740), $Rrs$(492)/$Rrs$(740)], and
green/red [$Rrs$(560)/$Rrs$(704), $Rrs$(665)/$Rrs$(704)] band ratios were significantly
correlated with *TSI* (Table 2). This result indicated that the blue/red and green/red band
ratios were more sensitive to the *TSI*, although the nutrients and SDD had no optical
response. It was known for decades that the blue part of spectrum is useless when water
itself is not blue (i.e. outside of ocean or very oligotrophic mountain lakes), owing to
the noneffective atmospheric correction and complex reflectance signals. However, our
dataset to train *TSI* models contain the samples from blue and oligotrophic Tibetan lakes,
which are like the oceanic environments (Liu et al., 2021). The blue bands responding
to *TSI* were thus used in this study. Most empirical Chl-a estimation studies adopted
red/near infrared (NIR) band ratios to calibrate models using reflectance signatures
(Gitelson et al., 1992). Similarly, empirical SDD retrieval models provided by previous
studies used empirical algorithms or models to figure out what bands should work the
best considered the following ratios: blue/green, red/blue plus red/green, and red/blue
plus blue (Bindling et al., 2007), and Red/Blue ratio plus Blue (Kloiber et al., 2002).
Kutser et al. (1995) also built a TP retrieval model using the red and NIR ratios, which
is consistent with Chl-a empirical models. Overall, it is not surprising for our *TSI* model
to have strong correlations with the blue/red and green/red band ratios because the *TSI*
incorporates the optical properties.
For this reason, we used MSI bands in the visible band ratios at six bands,
considering the comprehensive spectrum information about the trophic status of lakes as
input variables (Fig. 2). The three representative machine learning *TSI* models improved
the accuracy of the traditional linear regression (Table 2 and Fig. 5), and the results
were better than those obtained with B1-B6 reflectances as input variables (Fig. S1). As





a type of supervised machine learning algorithm, linear regression can be used to obtain
certain learning criteria as expressions ($y=w_0+w_1\times x_1+\ldots+w_p\times x_p$) about the optimal $w_i$
solution. However, for complex targeted tasks, the fitting ability of linear regression is
limited, and it cannot represent the real situation well. For example, a support vector
machine can map data to another space, which can use a linear regression to distinguish
the categories well. In complex environments (real world in machine learning), such as
our large-scale database collected from different lakes (Fig. 1), there are various
environmental factors as well as different seasons within a lake, that have an impact on
the trophic parameters and optical characteristics of lakes. Likewise, we found that the
enhanced input variables, like the band ratios, if appropriately corrected for the *TSI*,
resulted in a better performance (Fig. S1). This is consistent with some applications of
machine learning algorithms (Cao et al., 2020), in which the performance of machine
learning was reduced when covariances of input features were incorporated. This allows
us to find more interesting *TSI*-correlated band ratios for MSI imagery in machine
learning.

482         Several machine learning algorithms generally have different advantages and

applicability owing to their different main principles (Cao et al., 2020; Li et al., 2021).
This can be found in our results of the validation exercise, which showed that XGBoost
provided stable *TSI* estimates, with a slope close to 1 and a good fitting coefficient of
the measured and derived values ($R^2=0.87$, slope=0.85, MAE= 3.15, RMSE=4.11) (Fig.
4). Similarly, we can also find an excellent performance ($R^2=0.85$, slope=0.84,
MAE=3.31, RMSE=4.34) for estimating *TSI* values by the random forest algorithm.
This was likely because it is a summation of all weak learners, weighted by the native
log odds of error. In the case of boosting, we make decision trees into weak learners by
allowing every tree to make only one decision before prediction. In some cases,

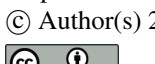



XGBoost outperformed random forest. In addition, the support vector machine
performed worse than XGBoost and random forest (Fig. 4). Li et al. (2021) used a
support vector machine to estimate Chl-a concentrations with a relatively small dataset
of 32 samples and 273 samples, respectively. This is consistent with the recent process
in the development of support vector machines and has many advantages for remote
sensing applications with a small number of training datasets. Overall, the remote
sensing and machine learning-based *TSI* model aims to reduce the dependence of
traditional field measurements, while also providing a cost-effective approach to rapidly
quantify the trophic state.
**4.2 *TSI* model for lake classifications**
We validated the XGBoost *TSI* model considering different scenarios of lake
classification, for example, water quality, optical absorption contributions, and
reflectance spectra (Figs. 2 and 6). The results indicate three application scenarios for
our model with low errors. The first one is of the XGBoost *TSI* model, which in
particular, performed well (slope>0.91, $R^2$>0.91) in high DOC (>3.88 mg $L^{-1}$) and EC
(>1000 µS $cm^{-1}$) lakes (Fig. 6). We found that lakes with high EC level correspondingly
showed a high DOC level (Table 1), for example, high average EC value of 5156.02 µS
$cm^{-1}$ and high average DOC value of 18.75 mg $L^{-1}$ for NAP-type lakes. These brackish
or saline lakes were distributed in the Tibet-Qinghai Plateau Region (e.g., KLK20, TS21,
QHH22, SLC32, BMC34, ZRNMC36, NMC37) and Inner Mongolia-Xinjiang Plateau
Limnetic Region (e.g., DL8, HSH10, DH17, HL18, WLSH16) (Table S1). Our results
are in agreement with those of previous studies that DOC and EC of inland waters
located in semi-arid region can be attributed to the evapo-concentration and
accumulation processes (Curtis and Adams, 1995) as well as anthropogenic activities.
Further, it can be observed that oligotrophic lakes accounting for 11.1% were also



distributed in the Tibet-Qinghai (Fig. 4).
Secondly, we found that our XGBoost *TSI* model performed well if the trophic
parameters that correlated to the $TSI_M$(Chl-a) or $TSI_M$(SDD) dominated the lake
classifications. Specifically, the high Chl-a (averaged 14.26 μg L$^{-1}$) and $a_{ph}$(440)
(averaged 0.26 m$^{-1}$) levels in NAP-type lakes showed the best performance (slope=0.98,
R$^2$=0.88) than those of other optical absorption contribution classifications (Fig. 6). In
fact, there was a negligible difference in the performance for application in Phy-type
and NAP-type lakes. For the third scenario, for the reflectance spectra classification,
cluster-1 lakes with low TSM (averaged 5.76 mg L$^{-1}$), turbidity (averaged 4.46 NTU),
and $a_d$(440) (averaged 0.26 m$^{-1}$) level, and high SDD level (average 2.38 m) also
showed good performance (slope=0.91, R$^2$=0.87) (Fig. 6). In general, *TSI*, as a
comprehensive index incorporating the optical properties of itself, was calculated using
trophic state parameters [($TSI_M$(Chl-a), ($TSI_M$(SDD), and $TSI_M$(TP) in Eq. 7]. Our
XGBoost *TSI* model performed best in the present study, which confirmed that the
performance was mostly determined by biogeochemical environments in larger-scale
regions. We cannot explain the dependence of the *TSI* model on the physico-optical
properties. From another point of view, it can be inferred that the XGBoost *TSI* model
applications mostly correlated to the Chl-a and SDD because of their high weight
allocation in *TSI* equation.

**4.3 Trophic status in five limnetic regions**
According to this study more than 50% of lakes were eutrophic, indicating a
long-standing status of eutrophication (Fig. 4), as seen by the mapping of 555 lakes by
our XGBoost *TSI* model (Fig. 7). Some lake investigations undertaken earlier in China
during 1978–1980 concluded that 41.2% lakes of eutrophication in China (Jin, 2003),





during 1988-1992 demonstrated that 51.2% lakes (Wang & Dou, 1998), during
2001-2005 indicated that 84.5% lakes, during 2011-2019 showed that 50% lakes (Wen
et al., 2019) were eutrophic or undergoing eutrophication. In our study, some historical
records of Chl-a, SDD and TP from in comparison to earlier national investigation by
Wang and Dou (1998) were collected in typical lakes, e.g., Dongting Lake, Poyang
Lake, Chaohu Lake, Taihu Lake and Jingpo Lake, respectively (Table S5). Evidently,
Chinese lakes have deteriorated considerably in terms of water quality at an alarming
rate for typical lakes, e.g., Jingpo Lake, Dongting Lake and Poyang Lake, during past
~22 years (Table S5). Lake eutrophication is influenced by both natural (hydrological
processes, topography, lake depth, and buffer capacity) factors as well as anthropogenic
factors (land-use changes, urbanization construction, and domestic and industrial
pollution) (Müller et al., 1998). A large-scale overview of lake eutrophication indicated
there was a significant difference (ANOVA, F=255.2, $p<0.001$) in the five limnetic
regions (Wang & Dou 1998). Owing to the imbalanced development of economic
(Fig.S2, GDP and population), geological topography (Fig.S3, solar radiation intensity
and sunshine hours) and climate (Fig.S4, annual temperature and precipitation), it was
not surprising that the eutrophic lakes were generally distributed in the Eastern Plain
Limnetic Region and Northeast Plain Limnetic Region, as well as that the oligotrophic
lakes were found in the Tibet-Qinghai Plateau Limnetic Region (Fig.4 and Fig.7).

561        Considering the natural factors for the distributions of Chinese lake eutrophication,

we could suppose some possibility that lake depth and lake hydrological processes
cause the eutrophication of lakes in China. Previous studies (Wang & Dou 1998; Huang
et al., 2014) have demonstrated that lakes with mean depths > 5 m in China are mainly
located in the Yungui Plateau Limnetic Region, Inner Mongolia-Xinjiang Plateau
Limnetic Region, and Tibet-Qinghai Plateau Limnetic Region, whereas almost all lakes



located in the Eastern Plain Limnetic Region are shallow. Both these lakes in the
Eastern Plain Limnetic Region are hydraulically connected with the Yangtze River with
a temporary residence time of approximately 30 days (Fig. S7). In shallow lakes, due to
wind waves or disturbance by fishes, the phosphorus/nitrogen nutrients stored in the
sediment can be easily resuspended and released into the overlying water (Niemistö et
al., 2008). Consequently, an increased frequency of algal blooms can be found in
Eastern Plain Limnetic Region, in lakes, such as Taihu, Chaohu, and Hongze (Qin et al.,
2019; Yao et al., 2016). Instead, deeper lakes, such as the ones in YGPLR and TQPLR,
possess relatively good buffer capacity for waste-water runoff (Huang et al., 2014).
Carvalho et al. (2009) found that Chl-a levels decreased with lake water depth and
geographic location. Qin et al. (2020) and Tong et al., (2006) demonstrated that
phosphorus reduction can mitigate eutrophication in deep lakes, and more efforts to
reduce both N and P need to be undertaken in shallow lakes. This can be demonstrated
in our case of Fuxian Lake with changeable eutrophication levels, with an average depth
of 87 m, which was the deepest lake in southwest China (Fig. S7). In addition, the
annual precipitation and air temperatures were relatively high in the EPLR (Fig. S4).
Hydrological and meteorological processes can scour land surfaces and bring nutrients
into lakes via rivers. Therefore, lake ecosystems were strongly related to the lake basin
morphology and its hydrologic characteristics, which were higher in shallow lakes than
in deep ones (Köiv et al., 2011).
On the other hand, human-induced eutrophication, for example, agricultural
fertilization (Carpenter, 2008; Huang et al., 2017), aquaculture (Guo & Li, 2003) and
sewage discharge (Paerl et al., 2011), are increasing terrestrial nutrient phosphorus but
not nitrogen concentration inputs (Schindler et al., 2008). We suspected that two
interactive factors, such as land-use and nutrient variations cause lake eutrophication,

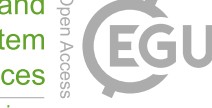

because this can be found in our investigation of distributed lakes in the EPLR in
comparison to earlier national investigation by Wang and Dou (1998). Many lakes in the
EPLR that were naturally connected with rivers have been modified to paddy fields, and
some small lakes have become isolated for lake aquaculture. For instance, Lake
Dongting was artificially shifted from being river-fed to dammed/isolated. Logically it
should a dam can settle down the suspended matter and nutrients via river inputs. But
the shallow characteristic and wind mixing influence process significantly increased the
probability of eutrophication (Liu et al., 2019). In EPLR and NPLR, 94% of China's
population lives in 43% of its eastern region, which visually demonstrates the
distribution of GDP with a densely populated east (Fig. S2). Owing to the requirements
of water source utilization, the EPLR has lost one-third of its original lake areas to
cropland since 1949 (Yin and Li, 2001). Lake aquaculture is highly active in these areas.
These processes could lead to terrestrial nutrient loading into lakes, from either
agriculture or aquaculture, and thereby alter the trophic state levels of a lake ecosystem.
In 2019, the total fish catch was 4,695,432, 25,588,135, 2,314,603, and 4,841,159 tons
in Hubei, Jiangxi, Anhui, and Jiangsu in the east, respectively (China rural statistical
yearbook).
Although we have not systematically analyzed the effects of environmental factors
on trophic status, some of the sparse existing comparative literature supported certain
spatiotemporal patterns. It should be emphasized that China has been facing serious lake
eutrophication and unbalanced distributions. Almost invariably, lake ecosystem health
would still be impacted by stresses integrating anthropogenic and overexploitation of
catchment resources. Consequently, addressing the issue of worsening eutrophication
requires a better understanding of the environmental interactive mechanisms in the
future.



## 5 Conclusions

Our study presents a novel remote sensing- and machine-learning-based algorithm applied in that allow to retrieve the lake TSI from Sentinel-2 MSI imagery. We used a match-up database ($N$=431) over a diverse range of bio-optical regimes to train machine learning algorithms and validated it against the in situ data. The trophic states of 555 lakes were then evaluated. These results provide a better understanding how remote sensing and machine learning-based models allow to estimate eutrophication over a large scale of different lakes. Our main findings can be summarized as follows:

1) Linear regression enabled us to find certain band combinations sensitive to *TSI* ($R^2$>0.59), for example, the blue/red [$Rrs$(443)/$Rrs$(740), $Rrs$(492)/$Rrs$(740)] and green/red [$Rrs$(560)/$Rrs$(704), $Rrs$(665)/$Rrs$(704)] band ratios.

2) XGBoost algorithm resulted in optimum performance with $R^2$=0.87 and slope=0.85, considering the low errors (MAE=3.15, RMSE=4.11), compared to the support vector machine and random forest algorithms.

3) If there is some preliminary data available from the study area one can improve the performance of the machine learning by dividing the lakes based on high DOC/EC, NAP-type and Phy-type, and cluster-1 reflectance spectra.

4) The trophic states of 555 lakes were evaluated in five limnetic regions; eutrophic lakes dominated in Eastern Plain Limnetic Region and Northeast Plain Limnetic Region, and most lakes in Tibet-Qinghai Plateau Limnetic Region were mesotrophic or oligotrophic.

In our subsequent research and management, qualification and mapping of *TSI* will be implemented as a remote sensing and machine learning model in a large-scale study, allowing for an improved performance. In the future, Sentinel-2 MSI data could be used to reveal spatiotemporal variations in lake trophic states in long-term time-series



642 responding to climate and anthropogenic activities.

643

## CRediT authorship contribution statement

Sijia Li: Conceptualization, Methodology, Formal analysis, Visualization, Funding acquisition, Writing original draft. Kaishan Song: Resources, Supervision, Project administration, Funding acquisition, Writing-review & editing. Tiit Kuster: Writing-review & editing. Ge Liu: Resources, Writing-review & editing. Shiqi Xu: Methodology. Zhidan Wen: Resources, Writing-review & editing. Yingxin Shang: Resources, Writing-review & editing. Lili Lyu: Investigation & Resources. Hui Tao: Investigation & Resources.

## Acknowledgements

The research was jointly supported by the China postdoctoral science foundation (2020M681056), the "Young support talents program" from Science and Technology Association of Jilin Province (2020-2023) to Dr. Sijia Li (QT202017), the National Natural Science Foundation of China (4217011915) and the Environmental Protection Project of Jilin Provincial Ecology and Environment Department (Nos. 2020-18). The authors thank all staff and students for their persistent assistance with both field sampling and laboratory analysis.

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
