# Peer review of "Remote Quantification of the Trophic Status of Chinese Lakes"

_Hydrology and Earth System Sciences, 2022_

## Author Comment (AC1)

Dear Editor,

Thank you for your letter and the reviewer comments concerning our manuscript entitled 'Remote Quantification of the Trophic Status of Chinese Lakes' (*hess-2022-91*).

We have studied comments carefully and have made correction which we hope meet with approval. Revised portion are marked in blue in the revised manuscript. The main corrections and the responses to the reviewer comments are described below.

Thanks again for your time and help.

Yours sincerely

Sijia Li and Zhidan Wen on behalf of the authors.

**Responses to Yumei Li' Comments**

Dear Yumei Li,

We would like to express our sincere appreciation for your careful reading and helpful comments. These comments are all valuable and helpful for revising and improving our paper, as well as the important guiding significance to our researches. We have studied comments carefully and addressed the points noted below.

Revised portion are marked in blue in the paper. The main corrections in the paper and the responds to the reviewer's comments and remarks are as flowing.

" Yumei Li: Peer review

The TSI is a universal paradigm for eutrophic research in scientific literature. It is very important to study how to quickly quantify tropical state index estimation in inland water, instead of traditional methods by deriving chlorophyll-a or clarity. The manuscript entitled "Remote Quantification of the Trophic Status of Chinese Lake" proposed an applicable machine learning algorithm which integrates a broad scale dataset of lake biogeochemical characteristics using Sentinel-2 Multispectral Imager (MSI) imagery. Authors applied the best one to first map 10-m TSI in 555 lakes across five limnetic regions in China, and comparison was conducted with previous investigation in lakes across China.

Overall, this paper is well organized and the logic is relatively clear. Many of the acquired datasets are also valuable. However, sufficient explanation is required for the following points.

- The Case 2 Regional Coast Color processor (C2RCC) was used to remove atmospheric effects. And In Figure 2, normalized water leaving reflectance of samples are acquired. How is the normalization done? This measurement is really odd. Then C2RCC-nets were used instead of C2RCC, and it is need to explain. There are some good atmospheric correction methods designing for MSI sensor, such as the Sen2Cor. Why you choose C2RCC for correcting atmospheric correction?

Response: Thanks for your patient review. The atmospheric correction (AC) process is to compensate the water-leaving reflectance signals and remove the contribution of atmospheric effect. The results can be found in our published paper Li et al., (2021, *science of the total environment*). The machine-learning C2RCC processor is built upon previous AC database of radiative transfer simulations and related TOA, relying on a per-pixel artificial neural network. It has two models with different application circumstances for open oceanic (nets) and inland waters (C2X). However, our published study on the comparison of different AC processors for lakes demonstrated that C2RCC can work better than Sen2Cor, iCOR, Polymer and Acolite. These in situ data were collected in four European lakes-Swedish Erken, Italian Garda, Estonian Saadjärv and Võrtsjärv, and Chinese typical lakes (2021, *science of the total environment*). The results showed that C2RCC-nets achieved quite good agreement

with high R² and slope closed to unity for Band 1-6 of MSI.

In addition, in order to evaluate the performance of ACs for inland turbid lake, we we collected lake samples of Chagan Lake (CG1) in spring (May 26th), august (July 18th and August 18th) and autumn (September 17th) in 2021. Chagan Lake has high variability in terms of suspended matter, algal abundance and trophic status. All measurements in these lakes were synchronized with satellite overpasses; we aim at to test against in situ $R_{rs}$ to find the best performing AC processors, based on the errors and R². Considering all bands, C2RCC-C2X and SeaDas processors performed better than those of other ACs, with C2RCC-nets had relatively low errors. Hence, considering the principles (Artificial neural networks, Hydrolight and NOMAD data) and performances in productive Chagan Lake, C2RCC was used to generate the TSI maps in this study. We hope that these revisions and the improved text will be satisfactory.

[Figure]

Figure the sample distirbutions of Chagan Lake (a) in 2021 with continuous Chl-a (b), TSM (c) and CDOM (d) records.

[Figure]

Figure Comparison of in situ measured reflectance spectra versus MSI reflectance spectra (Band1 to Band 6) of different ACs (a, Sen2Cor; b, C2RCC-nets; c, C2RCC-C2X; d, iCOR; e, Acolite; f, Polymer; g, SeaDAS and h, 6S) and their errors (i) in Chagan Lake.

- Several machine learning algorithms were used in the process of calibrating the TSI model. Why were these algorithms tested? Is there any reason? Is linear regression a machine learning approach?

Response: Thanks for your valuable comments. The reason is that RF, SVM and XGBoost are three typical machine learning algorithms. RF is a simple and intuitive

machine learning algorithm. The implementation of SVM is based on space conversion. Techniques and workarounds such as the kernel trick is used to solve the inseparability problem by introducing variables in SVM into a higher space and predict. XGBoost is an improved algorithm from gradient-enhanced decision tree (GBDT). It could train a set of weak decision tree model, and make each model to predict incorrect variables of previous model. In addition, the three machine learning algorithms are most widely used in remote sensing and environment field (Mountrakis et al., 2011; Toming et al., 2020; Cao et al., 2020).

Supervised machine learning is the construction of algorithms that are able to produce general patterns and hypotheses by using externally supplied instances to predict the fate of future instances. In short, supervised machine learning uses samples data to learn a model, and predict by test samples data.A hypothetical model is selected for the input sample space $x_i$, and the truth value $y_i$ is continuously fitted with certain learning criteria, and finally get the training model. For linear regression, the input space is $x_i$, the model expression is $y=w_0+w_1\times x_1+w_2\times x_2+\ldots+w_p\times x_p$ and the hypothesis space is $w_i$. Hence, LR is a kind of machine learning algorithms. We hope that these revisions and the improved text will be satisfactory.

- The water qualities optical absorption contribution and a k-mean clustering were used in this paper. How can these methods help with the search or the improvement of TSI algorithm? What are the advantages lies in? Please add more explanations.
Response: Thanks for your instructive comments and suggestions. The dataset used to develop the XGBoost covered a wide range of water qualities, optical absorption contribution and reflectance spectra. However, for signal lake, if the XGBoost could show excellent performance is required to evaluate. Hence, the *in situ* measured samples were classified in three ways, and XGBoost *TSI* algorithm was evaluated. If there is some preliminary data available from the study area one can improve the performance of the machine learning in calibration processes in future.

We are sorry to show the unclear explanation. For example, the XGBoost performed well in high DOC/EC lake scenario, high phytoplankton dominated NAP-type lakes scenario and cluster-1 lakes with low TSM scenario, respectively. Overall, in future work, for lakes mainly located in high elevation and arid region with high DOC/EC levels, the more input variables responding to CDOM (Green/Red) could be added in XGBoost TSI model. More classifications based on reflectance spectra (Spyrakos et al., 2018) and water color index (Wang et al., 2018) should be first used and then developed corresponding models for high turbid lakes. According to your suggestion, these explanations are added in revised manuscript (Section 4.2). We hope that these revisions and the improved text will be satisfactory.

-The time window of match-up dataset was chosen as 7 day. This time window might be too wide considering the dynamic change of water qualities. Further, it may not show the advantage of relatively high temporal resolution of MSI. And the credibility and accuracy could be undermined by the wide time window.
Response: Thank you for your comments. We agreed with your comment, the

time-differences between in situ data collection and satellite image acquisition could affect results because the water properties may change, for example due to heavy rain or emerging algal bloom that change water properties in hours rather than days. The MSI revisit time in many parts of the world is 2-3 days. For higher latitudes countries (Sweden, Finland and Estonia), the revisit time of MSI is almost every second day due to the overlapping orbits (Toming et al., 2016). In mid latitudes countries (Italy), there is also revisit time 2-3 days (Pereira-Sandoval et al., 2019). However, China stretches over 135°-73°E and 53°-3° N, by way of example in DaGuangBa (Table 1, 18°58'N,109°00'E), there are only 75 scenes level 1C products per year due to less orbits, in comparison to 175 scenes per year in Võrtsjärv (Estonia, 26°1'E, 58°15'N). One MSI has 10 days revisit time at the equator. There are two Sentinel-2's in space and the orbits overlap closer to the poles. We can expect that the results obtained there did not suffer from the time difference between in situ sampling and satellite overpasses. However, frequent cloud cover in some areas often prevents getting suitable imagers for several weeks in a row. Toming et al., (2016) and Cardille et al., (2013) found that in some cases even longer (weeks to month scale) time differences may still be acceptable. Secondly, many lakes, like Nam Co in Tibet (>4000 m), are hard to reach and the Sentinel-2 overpasses there are not as frequent as in many other places. Previous studies (Olmanson et al., 2008; Song et al., 2020) demonstrated that ±7 days time window that has been shown to be adequate for derivation of SDD using satellite imagery. We hope that these revisions and the improved text will be satisfactory.

-Quantitatively, this article referred an effective way to monitor the lake eutrophication on a macro-scale. The machine learning seems to be more excellent than traditional empirical models. However, it may be not a discontinuous mapping within a lake in supplementary material.

 Response: The authors really thank for your instructive comments. We think that a discontinuous mapping within the same lake may related to the influence of cloud. In our revised manuscript, we corrected these images in supplementary material. We hope that these revisions and the improved text will be satisfactory.

[Figure]

Special comment

-Line 43 knowledge of the process of eutrophication can provide us with an understanding of the …... is confused, please clarify it.

Response: Thank you for your patient review. This sentence was corrected as '*Hence, knowledge of eutrophication process can provide us with an understanding of the structure and function of lake ecosystems that give rise to environmental changes*'. (Line 42-44). We hope that these revisions and the improved text will be satisfactory.

-Line 158 some lakes were sampled in the middle can be described again.

Response: Thanks for your comment. The surface areas of some lakes are so small such as TPC (1) and Xingxingshao (2), with surface area is 25.86 km$^2$ and 8.96 km$^2$ respectively. This indicated that there were no significant spatial patterns of water qualities in small-size waters. In fieldwork, small-size waters were sampled in the middle. Conversely, Qinghai Lake, seen as the largest saline lake in China, was revisited many times, and more water samples ($N$=32) were collected at multiple locations evenly distributed over the lake. We revised this sentence (Line 157-160) according to your suggestion. We hope that these revisions and the improved text will be satisfactory.

-Line 204-205 total phosphorus did not show optical properties, but it still appeared in the modified TSI calculation. Is it possible to explain again?

Response: Thank you for your patient review. Carlson, (1997) reported that improved TSI was calculated by Chl-a, SDD and TP. Nutrients as one of the main driving factors to phytoplankton growing and photosynthesis. However, for water color remote sensing, Chl-a (algal absorption at blue and red wavelength) was one of optical active substances (non-algal particle and CDOM), when pure water is a constant. Although the nutrients and SDD had no optical response, they still indirectly have impact on lake eutrophication. We hope that these revisions and the improved text will be satisfactory.

-Line 218 I am not sure that the selection of images with time window ±7 days can affect the reflectance and results because of quick changes of water qualities, such as a storm event.

Response: Thank you for your patient review. Our response to the selection of images with time window ±7 days can be found in above comments. We hope that these revisions and the improved text will be satisfactory.

-Line 234 why the four algorithms used in this study are the representative machine learning algorithms?

Response: Thanks for your comment. Our response to four algorithms used in this study can be found in above comments. We hope that these revisions and the improved text will be satisfactory.

-Line 283 this section needs to be improved and one or two sentences are included

Response: According to your suggestion, we improved this section (Line284-286). We hope that these revisions and the improved text will be satisfactory.

-Line 447-451 need to be improved. It seems the blue band is useless in some high turbid or productive waters, but it is included in this study owing to some samples from Tibet.

Response: Authors really thank you for your suggestion. Our dataset were both collected from turbid lakes and clear Tibet lakes. Although blue band is useless in lakes with abundant non-algal and suspended matter, the performance of our model used blue band as input variables. This is because the TSI is a comprehensive index, of which the SDD and TP are not the optical active substances. Then the our dataset to train *TSI* models contain the samples from blue and oligotrophic Tibetan lakes, which are like the oceanic environments. The blue band can be used in our model. We hope that these revisions and the improved text will be satisfactory.

Technical
- Line 102 Sentine-2 instead of Sentinel-2
Response: We are sorry for our careless mistake. Sentine-2 was corrected in revised manuscript.

- Line the same reference in Line 154 and 157 is different
Response: We are sorry for our careless mistake. The reference is Song & Li et al., 2019.

- note that TSI in some sentences are italic, and some are not, such as Line 213 and 214, as well as the N in Figure 8.
Response: We are sorry for our careless mistake. TSI in Line 213 and 214 are italic. Then we check all TSI in revised manuscript.

- Many scripts (e. g., R2) require superscripts or subscripts for proper rendering, such as Figure 6a

Response: Thank you for your comment. We corrected these mistakes in our revised manuscript.

- some typefaces have different colors in Figure 7.

Response: Thank you for your comment. We corrected these mistakes in our revised manuscript.

- Line 577 Qin et al., (2020)

Response: Thank you for your comment. We corrected these mistakes in our revised manuscript.

- Line 606 it is very confused that there are many numbers and commas.

Response: We are sorry for our careless mistakes.    This sentence was corrected as '*In 2019, the total fish catch in Hubei was 4,695 tons; in Jiangxi was 432, 25 tons; in Anhui was 588,135 tons; 2,314,603 and 4,841,159 tons in Anhui and Jiangsu in the east, respectively (China rural statistical yearbook).*'

- Figure 2 CR2CC?

Response: We are sorry that this mistake was corrected. Then we checked these mistakes in revised manuscript.

---

## Author Comment (AC2)

Dear Editor,

Thank you for your letter and the reviewer comments concerning our manuscript entitled 'Remote Quantification of the Trophic Status of Chinese Lakes' (*hess-2022-91*).
We have studied comments carefully and have made correction which we hope meet with approval. Revised portion are marked in blue in the revised manuscript. The main corrections and the responses to the reviewer comments are described below.
Thanks again for your time and help.
Yours sincerely
Sijia Li and Zhidan Wen on behalf of the authors.

**Responses to Anonymou Referee#1' Comments**

Dear reviewer,

We would like to express our sincere appreciation for your careful reading and helpful comments. These comments are all valuable and helpful for revising and improving our paper, as well as the important guiding significance to our researches. We have studied comments carefully and addressed the points noted below.

Revised portion are marked in blue in the paper. The main corrections in the paper and the responds to the reviewer's comments and remarks are as flowing.

"In this manuscript, the authors present a methodological framework, using stepwise multiple regression analysis to find some band ratios and establish the XGBoost of machine learning approaches to estimate lakes TSI across China. Transferability and applications of XGBoost were tested in three different water classification scenarios, which provides a new idea for complex water color remote sensing modeling of class II water. This manuscript is well written and organized and this work is very meaningful, the method used is reasonable, and the model performance is good. I only have a few minor comments.

1. Line 37, 475, one or two references here can be helpful.
   Response: Thanks for your patient review. According to your suggestion, we add a reference (Guo et al., 2020; *Environmental Pollution,* Photo-induced phosphate release during sediment resuspension in shallow lakes: A potential positive feedback mechanism of eutrophication) in Line 37; a reference (Wen et al., 2016; *HESS,* Influence of environmental factors on spectral characteristic of chromophoric dissolved organic matter (CDOM) in Inner Mongolia Plateau, China). We hope that these revisions and the improved text will be satisfactory.

2. Line 339, 445, linear regression was used to identify significantly sensitive spectral variables related to TSI, the authors state that blue/red, green/red band ratios showed a good regression coefficient (R2 >0.59) with TSI. Have the results (R2 >0.59) been compared with other band combinations? It is best? What about other band ratios? The selection process of the optimum band should be described in detail. The tables or figures with comparative results should be given.
   Response: Thanks for your patient review. Linear regression was used to identify significantly sensitive band combinations could increase robustness of algorithms. According to you suggestions, we add some linear regression coefficients (2-tiailed) between band combinations and TSI in TableS4 (Supplementary Material). We hope that these revisions and the improved text will be satisfactory.

TableS4 The linear regression coefficients (2-tiailed) between band combinations and TSI

| Band combinations | $R^2$ | Band combinations | $R^2$ |
|---|---|---|---|
| Band 4/ Band 5 | 0.71[**] | Band 2+Band 3 | 0.54[**] |
| Band 3/ Band 5 | 0.68[**] | Band 2+Band 4 | 0.58[**] |
| Band 2/ Band 6 | 0.60[**] | Band 2+Band 6 | 0.50[**] |
| Band 1/ Band 6 | 0.59[**] | Band 3-Band 4 | 0.40[**] |
| Band 1/ Band4 | 0.47[**] | Band 3-Band 5 | 0.40[**] |
| Band 3/ Band 2 | 0.53[**] | Band 4-Band 5 | 0.50[**] |
| Band 6/ Band 5 | 0.58[**] | Band 4-Band 6 | 0.49[**] |
| Band 1-Band 3 | 0.48[**] | Band 5-Band 6 | 0.49[**] |

[**] presents the significance level <0.01

3. Line 382, What is the principle of selecting 555 representative lakes for mapping? Are the mapping images of 555 lakes consistent in date? As you know, the TSI derived from images of different seasons, cloud be completely different.

Response: Thanks for your patient review. Wang and Dou et al., (1998) conducted a national early investigation of lake water qualities, and grouped them into five limnetic regions. The results demonstrated that 51.2% lakes of eutrophication in China (Wang & Dou, 1998). Hence, in our study, some historical records of Chl-a, SDD and TP from in comparison to earlier national investigation by Wang and Dou (1998) can be collected and compared. Although some lakes disappeared referring to the numbers recently according to statistics from Ma et al. (2011), these typical Chinese lakes have deteriorated considerably in terms of water quality (calculated TSI) at an alarming rate (Table S6).These can be found in revised manuscript Line 553-562.

Thanks for your suggestion. We selected the MSI images and XGBoost algorithm to map our TSI considering cloud-free images in autumn (Sep. and Oct) with 7-day time window. Then some typical lakes were lengthen the time-window. This is because China stretches over 135°-73°E and 53°-3° N, by way of example in DaGuangBa (Table 1, 18°58'N,109°00'E), there are only 75 scenes level 1C products per year due to less orbits, in comparison to 175 scenes per year in Võrtsjärv (Estonia, 58°15'N, 26°1'E). For higher latitudes countries, the revisit time of MSI is almost every second day due to the overlapping orbits (Toming et al., 2016). There are few clouds in autumn, and some lakes in Tibet are covered by ice in June and July. Hence, most of good images were acquired in autumn to maintain synchronization (Olmanson et al., 2008; Song et al., 2020). The revised text can be found in Line 379 and 381. We hope that these revisions and the improved text will be satisfactory.

4. Line 484-493, the results showed the support vector machine performed worse than XGBoost and random forest. Why? I suggest that specific reasons need to

be explained clearly, from the mechanism of the algorithm or drawbacks or advantages.

Response: Thanks for your patient review. This is because RF and XGBoost are integrated algorithms based on multiple single trees and decision trees. Then the trees are unpruned and diverse, leading to Unpruned and diverse trees lead to a high resolution in the feature space. For continuous features, it means a smoother decision boundary. According to your suggestion, we added these in revised manuscript (Line 436-440). We hope that these revisions and the improved text will be satisfactory.

5. Line 490, the references should be cited here.

Response: Thanks for your patient review. According to your comment, we cited references in revised manuscript (Chen et al., 2016). We hope that these revisions and the improved text will be satisfactory.

Line 404, Fig.8 did not been cited in text.

Response: We are sorry for our careless mistakes. In revised manuscript, we corrected these (Line 391). We hope that these revisions and the improved text will be satisfactory.

---

## Author Comment (AC3)

Dear Editor,

Thank you for your letter and the reviewer comments concerning our manuscript entitled 'Remote Quantification of the Trophic Status of Chinese Lakes' (*hess-2022-91*).

We have studied comments carefully and have made correction which we hope meet with approval. Revised portion are marked in blue in the revised manuscript. The main corrections and the responses to the reviewer comments are described below.

Thanks again for your time and help.

Yours sincerely

Sijia Li and Zhidan Wen on behalf of the authors.

**Responses to Anonymous Referee #2' Comments**

Dear reviewer,

We would like to express our sincere appreciation for your careful reading and helpful comments. These comments are all valuable and helpful for revising and improving our paper, as well as the important guiding significance to our researches. We have studied comments carefully and addressed the points noted below.

Revised portion are marked in blue in the paper. The main corrections in the paper and the responds to the reviewer's comments and remarks are as flowing.

"Dear Authors,

The paper presents a framework to quantify remotely trophic states in lakes. The study was conducted in China, but the developed approach can be applied in other parts of the world for water quality management. The study is interesting and shows novelty. I have only a few comments:

Major comments:

1. The results on the spatio-temporal patterns of ungauged lakes are roughly described (section 3.5), although it is the end goal of the presented remote quantification approach. Please, describe the spatio-temporal patterns of trophic states in detail. Additionally: How long is the input time series used for the XGBoost model? Are there any trends of eutrophication? Also, the Authors claimed that trophic states in lakes are related to lake basin characteristics, climate, and anthropogenic activities. However, the presented results are not enough to support this affirmation. Please, compare basin characteristics, climate and anthropogenic activities to the predicted TSI.

Response: Thanks for your patient review. We are sorry our unobvious description about small title '3.5 Spatio-temporal patterns of trophic states in a large-scale overview' for the patterns of trophic states. This is because the MSI sensor equipped in Sentinel-2 was launched since 2015, and there are no images in early period like Landsat series satellites (1975-present) to provide the interannual variation of trophic state. The MSI revisit time in many parts of the world is 2-3 days. For higher latitudes countries (Sweden, Finland and Estonia), the revisit time of MSI is almost every second day due to the overlapping orbits (Toming et al., 2016). In mid latitudes countries (Italy), there is also revisit time 2-3 days (Pereira-Sandoval et al., 2019). However, China stretches over 135°-73°E and 53°-3° N, by way of example in DaGuangBa (Table 1, 18°58'N,109°00'E), there are only 75 scenes level 1C products per year due to less orbits, in comparison to 175 scenes per year in Võrtsjärv (Estonia, 26°1'E, 58°15'N). One MSI has 10 days revisit time at the equator. There are two Sentinel-2's in space and the orbits overlap closer to the poles. Secondly, many lakes, like Nam Co in Tibet (>4000 m), are hard to reach and the Sentinel-2 overpasses there are not as frequent as in many other places. In addition, frequent cloud cover in some areas often prevents getting suitable images for several weeks in a row. Hence, we

selected some representative lakes to qualify spatial trophic states with a total of 139 cloud-free images in spring (Apr. and May.), summer (Jul. and Aug.) and autumn (Sep. and Oct.) to analyze annual variations using our the XGBoost algorithm. According to your suggestion, we revised this title as '*3.5 Spatial and Seasonal patterns of trophic states: Five lake limnetic regions*'.

According to your suggestion, we revised manuscriptas following '*Previous studies have demonstrated that some lakes disappeared or increased numbers recently according to statistics from Ma et al. (2011). Thus, we selected some representative and stable lakes (N=555) to qualify spatial trophic states using the XGBoost algorithm. The preprocessing of MSI data were referred to the Fig.2, and a total of 139 cloud-free images in spring (Apr. and May.), summer (Jul. and Aug.) and autumn (Sep. and Oct.) covered investigated lakes were acquired. According to the different geographic and limnological types in China, lakes were divided into five limnetic regions (Wang and Dou 1998, Early National Investigation): Eastern Plain Limnetic Region (EPLR, N=123), Northeast Plain Limnetic Region (NPLR, N=37), Inner Mongolia-Xinjiang Plateau Limnetic Region (IMXPLR, N=56), Yungui Plateau Limnetic Region (YGPLR, N=15), and Tibet-Qinghai Plateau Limnetic Region (TQPLR, N=324) (Fig. 1 and Supplementary data).*
*In general, there were significant seasonal variations in eutrophic state for lakes from the EPLR (F=39.56, p<0.001) and TQPLR (F=5.0, p<0.05) (Fig. 7). The averaged TSI in EPLR were 56.37 (Spring), 57.73(summer) and 54.26 (autumn) indicating serious eutrophication of investigated lakes, consistent with the results from Li et al., (2022). Recognizing that over 94% of the Chinese population lives in eastern watersheds with great demands of water use, this may be due to different water qualities management in provincial scales. Likewise, we found there was spatial heterogeneity of TSI results in TQPLR and some of which were the widespread saline lakes in Qinghai-Tibet Plateau with high reflectance in satellite images. On the contrary, there were no seasonal differences of TSI for lakes from IMXPLR, NPLR and YPLR, respectively. The eutrophic lakes dominated the proportions of the investigated lakes in the EPLR (93.5 %), followed by the NPLR (89.2 %), YGPLR (86.7 %), IMXPLR (69.6%) and TQPLR (3.7%) (Fig.8). It can be also found that mesotrophic lakes were found in the decreased order of TQPLR (45.7 %), IMXPLR (30.4%), YGPLR (13.3 %), NPLR (10.8 %) and EPLR (6.5 %), respectively. In comparison, most oligotrophic lakes (50.6%) were distributed in the TQPLR.* '

We agreed with your comment, and we are sorry for our careless descriptions about 'trophic states in lakes are related to lake basin characteristics, climate, and anthropogenic activities'. In future work, we plan to produce long time series and large-scale trophic state products and analyze the driving forces results of basin characteristics, climate and anthropogenic activities. Our presented results are not enough to support this affirmation, and this sentence was deleted in revised manuscript. We hope that these revisions and the improved text will be satisfactory.

2. What are the limitations and uncertainties of this study? The limitations and uncertainties should be presented as a section in Discussion.
Response: Thanks for your patient review. According to your suggestion, we add the limitations and uncertainties of this study as a section in Discussion in revised manuscript as following

'*4.4 Limitations, uncertainties and future*

*Toward the United Nation's Sustainable Development Goal (SDG) 6.3.2, satellite imagery and machine learning still provides great potential for evaluating water qualities state from global observations, particularly in developing countries. Machine learning algorithms could serve as good alternatives for empirical and semi-analytical algorithms to quantify on large-scale spatial applications, which could avoid or minimize the errors. Our results further demonstrated machine learning algorithms could improve the accuracy of water quality models (e.g., TSI) when the linear regression was used to find sensitive band combinations with red/red edge bands. Previous studies (Li et al., 2021, 2022) found red and red edge band could help us to quantify the spatial and temporal changes of Chl-a concentration or a synthetic parameter-such as TSI with high Chl-a weight ratio-from regional lakes. It is enable us to use sentinel-2 or similar sensors equipped with these bands to capture records of TSI dynamics.*

*As a medium-resolution (10~60 m) satellite, Sentinel-2 MSI offers the potential to monitor small-size lakes and produce reliable TSI estimates. However, there are significant obstacles in generating a Sentinel-2 (~10m) lake TSI distribution, including the acquisition of high quality atmospheric corrected $Rrs(\lambda)$ and massive computational overhead by C2RCC processor (Li et al., 2023). C2RCC processor designed for waters based on neural networks is data-driven approach and uses huge datasets collected from in situ and simulation measurements. In situ reflectance measurements were not conducted in these investigated Chinese lakes when sampling. Our recently study reported that C2RCC (SNAP 8.0) and Polymer (v4.13) processors both performed best with in situ field radiometry in typical lakes across China (Li et al., 2023), but the latter could work better when all bands are pooled together in derived algorithms. Considering the growing requirements of TSI products, more in situ measurements would be required to be added the already-implemented processors in following work.*

*In addition, there is a need for a robust model developed from different locations and optical water types that accounts for the interplay of different water quality parameters. Machine learning TSI model required a highly calibrated dataset, including high nutrients (e.g., TP >2.50 mg $L^{-1}$ in this study) and Chl-a concentrations (>100 μg $L^{-1}$ in this study). Likewise, for our developed universal TSI model, the feasibility application performances were different considering lake classifications. Hence, the extensive field–lab materials with complex source variations would be required first and water optical typologies further is a good compromise to develop groups of optimized algorithms in future. Nevertheless, we aim to provide technical operation approach, which could prompt more analysis responding to warming climate and anthropogenic activities. The strong linkages between reflectance and several trophic state defining indexes further underscore the potential of remote sensing for resources-limited countries meet their SDG goals.*'

We hope that these revisions and the improved text will be satisfactory.

Minor comment:
1. Lines 100-101: "there are 117 million of lakes on Earth (Verpoorter et al. 2014)" This information was already provided before.
Response: We are sorry for our careless mistakes. In revised manuscript, we corrected these as following '

*Inland water TSI has been produced for large lakes using MODIS sensor (Wang et al 2018).*

*However, this study is for more than 2000 large lakes (due to the spatial resolution of the sensor).'.*

We hope that these revisions and the improved text will be satisfactory.